# The Interplay between Glioblastoma Cells and Tumor Microenvironment: New Perspectives for Early Diagnosis and Targeted Cancer Therapy

**DOI:** 10.3390/brainsci14040331

**Published:** 2024-03-29

**Authors:** Assunta Virtuoso, Giuseppa D’Amico, Federica Scalia, Ciro De Luca, Michele Papa, Grazia Maugeri, Velia D’Agata, Celeste Caruso Bavisotto, Agata Grazia D’Amico

**Affiliations:** 1Laboratory of Neuronal Networks Morphology and System Biology, Department of Mental and Physical Health and Preventive Medicine, University of Campania “Luigi Vanvitelli”, 80138 Naples, Italy; assunta.virtuoso@unicampania.it (A.V.); ciro.deluca@unicampania.it (C.D.L.); michele.papa@unicampania.it (M.P.); 2Department of Biomedicine, Neurosciences and Advanced Diagnostic (BiND), Human Anatomy Section, University of Palermo, 90127 Palermo, Italy; giuseppa.damico01@unipa.it (G.D.); federica.scalia02@unipa.it (F.S.); 3Department of Biomedical and Biotechnological Sciences, Section of Anatomy, Histology and Movement Sciences, University of Catania, 95100 Catania, Italy; graziamaugeri@unict.it (G.M.); vdagata@unict.it (V.D.); 4Euro-Mediterranean Institute of Science and Technology (IEMEST), 90139 Palermo, Italy; 5Department of Drug and Health Sciences, University of Catania, 95125 Catania, Italy; agata.damico@unict.it

**Keywords:** glioblastoma multiforme, tumor microenvironment, extracellular vesicles, chaperoning system

## Abstract

Glioblastoma multiforme (GBM) stands out as the most tremendous brain tumor, constituting 60% of primary brain cancers, accompanied by dismal survival rates. Despite advancements in research, therapeutic options remain limited to chemotherapy and surgery. GBM molecular heterogeneity, the intricate interaction with the tumor microenvironment (TME), and non-selective treatments contribute to the neoplastic relapse. Diagnostic challenges arise from GBM advanced-stage detection, necessitating the exploration of novel biomarkers for early diagnosis. Using data from the literature and a bioinformatic tool, the current manuscript delineates the molecular interplay between human GBM, astrocytes, and myeloid cells, underscoring selected protein pathways belonging to astroglia and myeloid lineage, which can be considered for targeted therapies. Moreover, the pivotal role of extracellular vesicles (EVs) in orchestrating a favorable microenvironment for cancer progression is highlighted, suggesting their utility in identifying biomarkers for GBM early diagnosis.

## 1. Introduction

Glioblastoma multiforme (GBM) is the most aggressive brain tumor that accounts for 60% of primary brain cancers, with a poor survival rate [1,2]. The survival time is usually less than one year after diagnosis, while for lower-grade gliomas, the survival time is longer [3]. In general, the 5-year survival rate of GBM patients remains around 10% [4]. Moreover, most patients with GBM do not have a family history, a characteristic shared by most central nervous system (CNS) tumors [5]. Despite the progress of research in areas such as bivalent chimeric antigen receptor (CAR) T cells and immunotherapies [6], chemotherapy and surgery represent the current therapeutic options. The gold-standard therapy in GBM treatment is Temozolomide (TMZ), which is usually administered in combination with radiotherapy for about six cycles. However, this therapeutic approach shows side effects including myelosuppression and hepatic impairment [3,7,8].

The onset of GBM is usually unpredictable and characterized by a poor prognosis. Therefore, its early diagnosis by tumor markers becomes crucial.

The high molecular heterogeneity of gliomas, the intricate relationship with the tumor microenvironment (TME), and the non-selective treatments available are often responsible for events of epilepsy and neoplasia recurrence [9].

Most GBM cases are diagnosed in the advanced stage, as the symptomatology arises from their rapid expansion due to the tumor’s infiltration towards healthy brain structures. Neuroimaging techniques, such as magnetic resonance imaging (MRI) or computed tomography (CT), show the GBM as a bulk with invasiveness paths that seem to have an anatomical fingerprint [5,10]. Recurrent GBM often arises in peritumoral areas. Therefore, the radiologically defined FLAIR hyperintensity areas beyond the GBM enhancing core becomes matters of debate. T2-flair digital subtraction maps may be useful tools for quantitation, and the surgical procedure of “flairectomy” could be of benefit to patients’ outcomes [11,12].

Beyond the biology of oncological transformation and its genetics, the aggressiveness of the GBM is allowed by the alliance with other cells in the central nervous system (CNS), including neurons, astrocytes, oligodendrocytes, macrophages, microglia, and other immune populations [13,14,15]. Therefore, the CNS microenvironment is indispensable for GBM progression [16].

Astrocytes and myeloid-derived cells including microglia and macrophages share innate immune properties and are the first interactors of GBM. The tumor-supportive glial cells aid the cancer’s immune escape and facilitate its progression in a dynamic sequence of molecular events [17]. In accord, many shreds of evidence indicate that GBM evolves by the reorganization of TME, suggesting the potential of glial and myeloid cells’ re-education as an immunotherapy approach and the urgency of novel biomarkers for the early diagnosis of the disease [18,19].

GBM cells establish contact with the host cells and can release cytokines or extracellular vesicles (EVs), changing their metabolism and converting them toward a pro-tumor phenotype [20].

In the current review, we depict the main molecular interactors between human GBM cells, astrocytes and microglia, supported by the literature and bioinformatic tools. We identify molecular targets specific to astrocytic and myeloid cells that could potentially be involved in cancer progression. Moreover, we highlight the role exerted by EVs in the early and targeted diagnosis of astrocytic tumors. 

### 1.1. Human GBM and Astrocytes

Human astrocytes are star-like cells, a major glial component in the CNS. They provide neuronal axon guidance during development and metabolic support to neurons throughout their life. They take part in synaptic activity, maintain ion homeostasis and the integrity of blood–brain barrier (BBB), and concur to build the extracellular matrix (ECM), along with many other roles that are as-yet undiscovered in health and disease. Human astrocytes can be identified for the expression of the glial fibrillary acidic protein (GFAP), which increases in response to injury [19,21], even though its labeling appears weak and diffused in the tumor area [22]. Human glioma cells may undergo an alteration in GFAP splicing, with a dominance of the GFAPδ isoform in GBM, affecting the aggressiveness of the tumor [23]. Reactive astrocytosis with typical overexpression of GFAP and vimentin (VIM) is commonly found at the boundary of the tumor bulk and in the peritumoral microenvironment [24,25]. Furthermore, VIM expression appears to be elevated in human glioblastoma cells compared to healthy astrocytes [26]. Glutamate transporter 1 (GLT-1) is expressed primarily in astrocytes [27]. GLT-1 mRNA is scarce in primary brain tumors when compared to control brain tissues, and it seems to be inversely correlated with the tumor grade [28]. Peritumoral reactive astrocytes show an impairment in glutamate and potassium uptake, as well as glutamine synthetase activity [29]. Accordingly, the increased expression of GLT-1 in peritumoral tissue is neuroprotective and associated with prolonged survival in a rat model of experimental malignant glioma [30].

A recent analysis of astrocyte transcriptomes shows that tumor-associated astrocytes undergo a robust modification of the RNA transcripts. Peritumor astrocytes downregulate the expression of genes involved in the synaptic circuits’ maturation, metabolism, glutamate transporters (*SLC1A2* and *SLC1A3*) and receptor sensing, which may limit their ability to respond to environmental stimuli, contributing to neuronal toxicity and epileptic discharges [25]. Human astrocytes appear with a reactive profile based on the upregulation of proinflammatory genes [25]. In parallel, tumor core astrocytes upregulate many anti-inflammatory genes via JAK-STAT and interferon-gamma response, contributing to an immunosuppressive microenvironment that facilitates tumor growth and infiltration [31].

Human astrocytes under the cancerous processes reveal a change in the protein secondary conformation, with less α-helical sequence content and an increase in β-sheets and random structures, critical to the biological function [24].

The phenotypic transformation of human astrocytes occurs via direct coupling with glioma cells. The astrocytic gap junction protein, Connexin 43 (Cx43), was identified in immunoreactive areas of cell–cell contact [32]. However, an inverse correlation has been revealed between Cx43 and the tumor grading in biopsies. The expression of Cx43 decreases as the glioma invasion increases and could be considered as a putative marker of glioma progression. Indeed, Cx43 labeling has been detected as weak, with aberrant cytoplasmic staining or fibrillary background in human glioblastomas [22]. Another study reported the persistence of elevated levels of Cx43 mRNA in high-grade astrocytomas. The reduced levels of the Cx43 protein could be due to an alteration in the post-transcriptional mechanisms such as the regulation of its synthesis and/or the intracellular transport to membrane sites [33]. From animal studies, it has been suggested that tumor cells contact the Cx43+ astrocytes to overcome the glial scar surrounding the tumor core [34]. Subsequently, glioma migration may depend on other mechanisms. Among these, human GBM cells release extracellular vesicles (EVs) which are taken up by other cells in the TME. EVs are nanoparticles with a lipidic membrane that contains various molecular constituents, including proteins and nucleic acids [35,36]. Among the exosome-associated markers, there are CD63, Tsg101, and ALIX, and an absence of GM130, a Golgi protein that indicates contamination with cellular debris [37].

GBM EVs appear to reprogram astrocyte metabolism by inducing a shift in gene expression that may be partly associated with the horizontal transfer of mRNAs encoding ribosomal proteins, oxidative phosphorylation, and glycolytic factors [38]. Human astrocytes receive EVs containing CD147, which is a signaling protein propagated from tumor cells to astrocytes inducing the expression and release of metalloproteinases 9 (MMP9), probably through the MAPK pathway [38]. Metalloproteinases are enzymes degrading the extracellular matrix (ECM), whose remodeling is a key process for glioma progression [37]. Beyond the MMPs, regulators of cell–cell or cell–substrate adhesion in the ECM such as collagen IV, CXCL14, TGFBI, FBLN5, ADAMTS2, TN-C, and CD44 are modified to facilitate tumor dissemination and can be mainly subscribed to microglia and astrocytes [39,40,41]. Reactive astrocytes maintain tumor pro-invasion programs driven by the signal transducer and activator of transcription 3 (STAT3), p53, and MYC signaling pathways. They aid tumor immunoevasion under the control of both GBM and the microglia [31,42,43].

### 1.2. Human GBM and Microglia

Infrared spectroscopy presented the human microglia as round, small, and short rod-shaped with clear edges [24]. Healthy, resting microglia, reactive microglia, intermediate and bumpy forms, and macrophage-like cells can be tagged by allograft inflammatory factor 1 (Iba1), CD68, CD16, and CD163 markers. They reveal different antigens with a specific localization: membrane associated and cytoplasm for Iba1, lysosomes for CD68, and cell membranes for CD16 and CD163. Iba1+ and CD16+ cells prevail in human low-grade gliomas, whereas CD68+ and CD163+ cells increase in high-grade gliomas, with a significant correlation between patients’ worsening overall survival [44]. Many other molecules, including growth factors and cytokines, have been identified as regulators of glioma-associated microglia cell interactions [45].

During cancer progression, microglia cells polarize in different phenotypes: the M1 phenotype induces reactive oxygen species (ROS) and nitric oxide (NO) and releases proinflammatory cytokines (IL-1 β, IL-6, TNF α, CCL2), resulting in anti-tumor effects [46,47,48]; the M2 phenotype regulates anti-inflammatory cytokine secretion (TGF-β and IL-10) as well as immunosuppressive factors (ARG-1 and CD36), leading to pro-tumor consequences [49,50,51]. Several studies have demonstrated that glioma-associated microglia show both phenotypes in human GBM specimens, and distinguishing between these two phenotypes is very difficult [52,53]. In the literature, the hypothesis that positive feedback exists between cancer cells and microglia has been largely supported. Cancer cells release inflammatory factors (CCL2, CXCL12, CX3CL1, GDNF, CSF-1), leading to microglia chemoattraction and eventually inducing its shift to the M2 phenotype. On the other hand, many factors released from microglia (STI1, EGF, IL-6, TGF-β) stimulate GBM cell migration and proliferation [53,54,55,56] (Figure 1).

Microglia were shown to internalize GBM-derived EVs, leading to their high proliferation rates and a shift in their cytokine profiles toward immune suppression [57]. These findings could be in accordance with the role of microglia as the major regulator of immune adaptation, which contributes to creating an immunosuppressive microenvironment functional to tumor evasion. The immune adaptation in microglia could involve mTORC1 signaling via the regulation of STAT3 and NfKB transcription factors following the crosstalk with GBM, since the human GBM-associated-microglia mTOR pathway and the poor infiltration of lymphocytes were positively correlated [58]. In agreement, in vitro studies reported that mTOR kinase inhibitors polarize glioma-activated microglia to express a pro-inflammatory profile [59], and efforts have been made to enable the brain-restricted inhibition of kinase targets [60].

### 1.3. Extracellular Biomarkers Involved in Early Diagnosis of GBM

GBM recurrence arises from cells at the invasive margin evading surgical debulking, yet the degree of similarity to bulk counterparts remains uncertain. Investigating the extracellular space and tumor matrix is crucial for comprehending tumor regrowth [61]. The tumor microenvironment, influenced by astrocytes and microglia supporting GBM invasiveness, presents challenges for developing treatments aiming to decrease recurrence rates and enhance overall patient survival. The brain’s ECM offers a unique invasion environment, and research suggests that M2-like microglia/macrophage polarization correlates with a poorer prognosis, while predominant M1 polarization indicates a more favorable prognosis in GBM patients [62]. Additionally, astrocytes, involved in blood–brain barrier modulation and neuronal plasticity, contribute to CNS damage and undergo astrogliosis, transforming into reactive astrocytes capable of tissue repair. Various pathways activate astrocytes into tumor-associated astrocytes (TAAs) during GBM growth, influencing modifications in the TME and the release of active mediators potentially useful as biomarkers for the early diagnosis and prognosis of GBM [63].

About 10 years ago, circulating tumor cells (CTCs) and circulating tumor DNA (ctDNA) were discovered. They received much attention as new biomarkers and introduced the concept of *liquid biopsy* [64,65,66]. The concept of liquid biopsy can be applied to all fluids in the human body (blood, urine, cerebrospinal fluid, saliva, etc.), and the main advantage is that it allows cancer screening and the prognosis and monitoring of the effectiveness of cancer therapies with minimally invasive techniques for the patient (e.g., blood sampling). Increased levels of specific proteins or nucleic acids in blood or other biological fluids reflect many pathological processes, including cancer [67]. Thus, liquid biopsy has advantages over classic tissue biopsy, which is a more invasive procedure and cannot be repeated as often as a liquid biopsy; the latter can be repeated several times to follow the evolution of the disease in its various stages and the response to anti-tumor therapy [68]. Another disadvantage of tissue biopsies is that they represent a small region of the tumor and, therefore, may not fully include the intra-tumor heterogeneity of GBM [69].

In GBM patients, plasma obtained after the centrifugation of blood or cerebrospinal fluid is usually used as a biological matrix for tumor markers. Tumor-specific circulating components include CTCs, ctDNA, miRNA, proteins, tumor-educated platelets (TEPs), and EVs [70,71,72]. CTCs are cells released into the bloodstream by the primary tumor or metastases. GBM-derived CTCs have been shown to have a phenotype like cancer stem cells, thus exhibiting characteristics such as resistance to radiation, chemotherapy, and stress-induced apoptosis [73]. Another characteristic of CTCs is that they undergo epithelial–mesenchymal transition (EMT), resulting in a more mesenchymal phenotype with greater migratory potential that underlies the process of metastasis formation [74]. The detection of circulating CTCs is correlated with the risk of recurrence after surgery [75].

The ctDNA comprises small DNA fragments (180–200 base pairs) released by cancer into the bloodstream, mainly from apoptotic cells. In particular, the presence of GBM-derived ctDNA in plasma is low, compared to other tumors, because the blood–brain barrier (BBB) limits its diffusion into the blood, and therefore, it is better to use cerebrospinal fluid (CSF) for this type of analysis [76]. Its importance consists of the ability to carry tumor-specific mutations [77]. It was seen that the glioma genome in the CSF contained a wide spectrum of genetic alterations and had similarities to the genome in tumor biopsies. Among the most common mutations found in gliomas both at the tumor biopsy level and at the ctDNA level in the patient’s CSF were mutations within the telomerase reverse transcriptase promoter (TERT), the protein-coding regions of TP53 and the catalytic domain of isocitrate dehydrogenase 1 (IDH1), CDKN2A/CDKN2B deletions, and epidermal growth factor receptor (EGFR) amplifications [78].

miRNAs are part of the non-coding RNAs and are involved in both physiological and pathological processes (e.g., cancer) through a mechanism of post-transcriptional gene expression regulation. Thus, they are implicated in the processes of proliferation, differentiation, invasiveness, and tumor metastasis formation [79]. GBM patients show the altered expression of several miRNAs in the bloodstream [80,81]. Circulating levels of miR-15b, miR-23a, miR-133a, miR-150, miR-197, miR-497, miR-548b, miR-128, and miR-342-3p were reduced in GBM patients compared to healthy controls; in contrast, miR-21, miR-221, miR-222, miR-210, miR-182, and miR-454 levels were upregulated in cancer patients [69]. These miRNAs play key roles in the proliferation, invasion, and angiogenesis of glioma cells, and their dysregulation is correlated with low survival rates. Therefore, they can be used both as prognostic and diagnostic biomarkers. Their usefulness as cancer biomarkers was confirmed by observing that normal levels of these miRNAs were restored after surgery and chemo-radiotherapy. Furthermore, it was seen that these miRNAs appear to be specific for gliomas, allowing for a differentiated diagnosis from other types of brain tumors [82].

The class of non-coding RNAs also includes long non-coding RNAs (lncRNA). Increased HOTAIR lncRNA in serum was associated with poor prognosis and early tumor recurrence in GBM. Conversely, GAS5 lncRNA upregulation is associated with a better prognosis and a reduced chance of recurrence [83,84].

Circulating protein markers of brain tumors include immunosuppressive acid protein, acid glycoprotein alpha-1, antitrypsin alpha-1, glycoprotein fibronectin, and thrombomodulin-1 [85]. Among the secreted proteins, some have been proposed as potential serum biomarkers of astrocytoma: soluble CD95, YKL-40, serum protein kinases, apolipoprotein E, cell adhesion molecules, and angiopoietin-1 and 2 [86]. Although the serum profiles for GBM identified through data mining exhibit a promising level of robustness, these findings warrant validation in a larger sample size. Such validation is essential to establish stronger correlations between individual markers and GBM, ensuring the sensitivity and specificity of circulating protein markers for GBM, both critical characteristics for effective biomarkers [86]. As stated above, the primary biological processes central to the development of gliomas involve the restructuring of the ECM and the occurrence of proneural–mesenchymal transition (PMT), in which cancer cell proliferation and invasion are closely tied to the proteolytic remodeling of the ECM. This process enhances the migratory potential of glioma cells by activating integrins and associated signaling pathways, ultimately leading to increased invasiveness and migratory capacity, which contribute to a dismal prognosis and resistance to conventional treatments. As gliomas progress towards malignancy, patients undergo various therapeutic interventions, resulting in the emergence of surviving cells with diverse phenotypic traits and resistance to radiation or chemotherapy. In this context, the elevated expression of proteins associated with proteotoxic stress, such as heat shock proteins (HSPs), is closely associated with poor prognosis and therapeutic resistance in gliomas [78]. HSPs promote tumor growth by stimulating cell proliferation and inhibiting cell death pathways. Additionally, HSPs exhibit chaperone activity for numerous proteins, including matrix-degrading enzymes involved in ECM degradation. Furthermore, HSPs play a crucial extracellular role in the invasive phase of metastasis by binding to matrix metalloproteinases (MMPs) [79]. The upregulation of HSP expression occurs in both stages of gliomagenesis and in the acquisition of chemo- or radio-resistant phenotypes. Consequently, HSPs represent potential targets for effective clinical strategies in the rational development of anti-glioma drugs.

Two classes of cytoplasmic proteins associated with cellular stress (HSP70) and neural stem cells (FABP7) have been proposed as biomarkers for GBM [86]. HSP70 is an anti-apoptotic chaperone with tumor-promoting activity that therefore shows higher concentrations in the plasma of subjects with GBM than in controls. It has been shown that when cells are under stress, they can express HSPs on the surface of the plasma membrane with the possibility of secreting them into the extracellular space. This is one of the mechanisms by which cancer cells modulate the tumor microenvironment, and it is also possible to identify the increased concentration of HSPs in circulation [87,88]. Furthermore, it was noted through in vitro models that HSPs can easily cross the BBB [89].

### 1.4. EVs Role in GBM Progression and Focus on HSP90/HIF/HO-1 Pathway

The new frontier for the analysis of tumor biomarkers by liquid biopsy is represented by EVs characterization. They are more representative of ctDNA, which reflects the information of apoptotic or dead tumor cells. Furthermore, EVs have a higher relative abundance in biological fluids than circulating tumor cells (CTCs), due to the ability of EVs to cross the blood–brain barrier [90,91,92].

EVs are a heterogeneous group of vesicles that can be classified according to their size, density, and mechanism of biogenesis into three main types: exosomes, microvesicles, and apoptotic bodies [93]. However, the International Society for Extracellular Vesicles (ISEV) encourages the use of the term “extracellular vesicles (EVs)” as a generic term for all secreted vesicles, considering the lack of consensus for the identification of specific markers to distinguish between the different subtypes of EVs [94].

They are produced by all the cells in the body and are involved in both normal and pathological cellular processes like intercellular communication and homeostasis. EVs allow their cargo (RNA, proteins, and lipids) to be transported from the parenteral cells to the recipient cells by traveling through the extracellular space, so they are present in all biological fluids and can be used in liquid biopsy [95]. The cargo vesicles are located within a phospholipid bilayer that protects them from enzymatic degradation, and this is one of the advantages of using EVs in the field of nanotechnology target therapy [96].

In pathological conditions such as cancer, EVs can promote the horizontal spread of malignancy by transporting their cargo into recipient cells, thus altering their physiological mechanisms and affecting their phenotype [90,91,92].

The interest that has grown in EVs in recent years for cancer research concerns their role in certain processes such as proliferation and invasion, drug resistance, and the polarization to a microenvironment favorable to cancer [97,98]. Other studies show that circulating EVs levels were significantly reduced post-operatively compared to pre-operative plasma samples, suggesting that a tumor causes increased EVs levels in patients’ plasma [99].

Among the proteins present in the cargo of EVs, proteins belonging to the Chaperoning System (CS), with most of them being HSPs, are found [87]. These proteins are necessary for the maintenance of cellular homeostasis and they are also involved in several stages of carcinogenesis [48,100,101,102,103]. In glioma, various heat shock proteins (Hsps) play significant roles in promoting malignant behavior. Specifically, HSP27, HSP60, and members of the HSP70 and HSP90 families are involved in processes like proliferation, migration, invasion, and tumor growth regulation. Additionally, these proteins contribute to cancer cell survival mechanisms, including resistance to apoptosis and adaptation to hypoxic conditions [104,105].

In the pathogenesis of gliomas, HSP90 plays a crucial role in metabolic rewiring and the transcriptional regulation of key genes involved in tumorigenesis and cancer progression [106]. HSP90 directly influences metabolic processes by regulating the activity and stability of various metabolic enzymes. Additionally, it indirectly affects metabolic networks and oncogenic pathways by modulating HSP90-dependent signaling pathways. Client HSP90 proteins are regulated by the ubiquitin/proteasome system among the mechanisms [107]. Cancer cells exploit HSP90’s chaperone function to protect mutated and overexpressed oncoproteins, such as hypoxic inducible factor (HIF) and vascular endothelial growth factor (VEGF), from misfolding and degradation, thereby enhancing cancer cell survival [108]. Within GBM, hypoxic microenvironments support the maintenance of stem-like cells known as glioblastoma stem-like cells (GSCs), where HSP90 is co-localized with stem cell markers and HIF [109]. Furthermore, HSP90 is implicated in GBM cell migration, invasiveness, and the regulation of survival and apoptosis pathways [108]. Remarkably, a recent study has elucidated that Hsp90 not only interacts directly with membranes but also plays a pivotal role in their deformation during the biogenesis of EVs, thereby being intricately linked to the dissemination of molecular mediators [110].

By inducing HIF transcription, hypoxia activates many downstream target genes including human heme oxygenase 1 (HO-1) [111,112,113]. Its overexpression was detected in many human cancers, where it exerts an antiapoptotic role through mitogen-activated protein kinase pathway induction, resulting in poor prognosis and chemoresistance. Noteworthily, HO-1 overexpression was detected in human gliomas as compared to non-malignant samples, suggesting its oncogenic role in GBM progression [114,115]. HO-1 was suggested as an angiogenic marker since it drives VEGF overexpression, promoting an aberrant neovascularization typical to GBM. Moreover, HO-1 aberrant levels were also linked to increased macrophage infiltration [116,117].

Therefore, with glioma cell-derived vesicles being implicated in tumor expansion and pro-angiogenic signaling [118], the expression of specific proteins, such as HSP90 and HO-1, present within EVs can be used as an early diagnostic biomarker after the isolation of vesicles from biological fluids [36,119,120].

## 2. Methods

Based on the aforementioned evidence, EVs appear to be fundamental in understanding glioma interplay with the microenvironment. However, they appear to be heterogeneous, like the several cell populations found in tumor tissue [10,121]. Profiling the EVs based on the cells of origin would provide insights into elucidating the molecular content and the interactions occurring between human glioma, astrocytes, and myeloid cells.

We examined the connections among proteins using the online tool STRING version 12.0 (www.string-db.org, accessed for the first time on March 10, 2023), which is an open-access, biological database and web resource of known and predicted protein–protein interactions.

We considered the most used markers for astrocytes and microglia/macrophages (listed in Table 1), as well as the major tags for EVs, as reported in the previous sections. We refined the most significant connections based on the STRING experimental datasets and well-known interactions. After that, we enriched the protein pathways including STRING-predicted interactions and cluster analysis. The enrichment proteins included regulators of the vesicular trafficking process (MVB12A, MVB12B, VPS37B, CHMP4A) and the dendritic cell nuclear protein 1 (DCANP1) of the astrocyte signature.

### New Perspectives for Early Diagnosis and Targeted Cancer Therapy

The STRING investigation resulted in specific connections between human GFAP+, VIM+ astrocytic proteins, and HSP90, which may be a central node in the human astrocytic EV traffic (protein pathway enrichment *p*-value: 3.75 × 10^−5^; Figure 2a). HSP90 is crucial to cancer cell growth and survival [106]. It is overexpressed in pathological reactive astrocytes (GFAP+, VIM+) to hold GLT-1 degradation [122], therefore providing a high level of glutamate for tumor progression and associated epilepsy.

The microglia/macrophage (Iba1+, CD68+) expression profile is linked to the HO-1 protein (protein pathway enrichment *p*-value: 0.00109; Figure 2b). HO-1 is a marker of neovascularization and macrophage infiltration in human gliomas [108] and has been reported as a novel potential therapeutic target to prevent osteopontin-dependent human glioma cell migration [114]. However, the STRING cluster investigation suggested that EVs might not prevail as a means of communication among myeloid cells. HO-1 is generally triggered by the generation of reactive oxygen and/or nitrogen species, which have an extremely short lifetime and quick effects that would be not compatible with the low-yield process of EVs [123,124,125].

The current data shed light on targeted pathways within glioma–glia crosstalk, paving the way for new perspectives on early diagnosis and molecular therapies. Interestingly, EV biomarkers of activated astrocytes, HSP90+, may be directly detected in the patient’s fluid, potentially informing on the disease state compared to controls [126]. Studies on human microglia–macrophage-derived EVs are still limited [127].

## 3. Conclusions

In conclusion, GBM presents a formidable challenge due to its molecular complexity and intimate association with the CNS microenvironment. Novel therapeutic avenues targeting glial and myeloid cell interactions, and leveraging EVs-mediated signaling, hold promise for improved outcomes. The identification of specific biomarkers within EVs cargo, such as HSP90, offers a potential avenue for non-invasive, early diagnosis on astrocytic tumor lineage [115,128]. The identification of HO-1 as a potential target for microglia/macrophage reactions sheds light on the development of targeted therapies, which may go beyond the EVs’ traffic (Figure 3). Further research into the intricate intercellular communication in GBM pathology is warranted to advance precision medicine approaches in combating this aggressive disease.

## Figures and Tables

**Figure 1 brainsci-14-00331-f001:**
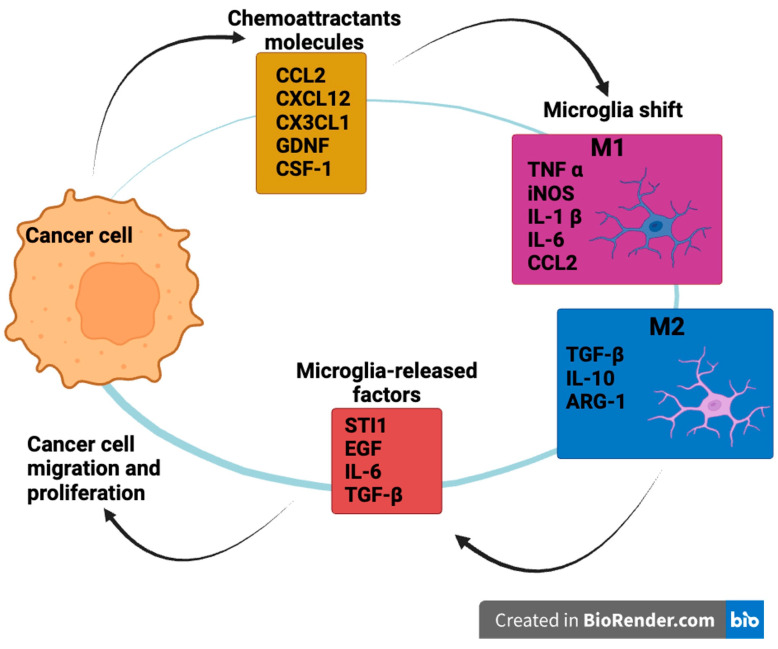
Positive feedback between cancer cells and microglia. Glioma cells release several chemoattractant molecules that induce microglia to shift from M1 (pro-inflammatory) to M2 (anti-inflammatory) phenotype. Among chemoattractants, there are monocyte chemotactic protein 1 (CCL2), CXCL12 (C-X-C Motif Chemokine Ligand 12), C-X3-C Motif Chemokine Ligand 1 (CX3CL1), glial cell-derived neurotrophic factor (GDNF), and colony-stimulating factor 1 (CSF-1). Activated microglia cells release many molecules that drive GBM cell migration and proliferation, including stress-inducible protein 1 (STI1), epidermal growth factor (EGF), transforming growth factor-β (TGF-β), and interleukin-6 (IL-6). (Image was created with BioRender.com online software: https://www.biorender.com. Accessed on 19 January 2024).

**Figure 2 brainsci-14-00331-f002:**
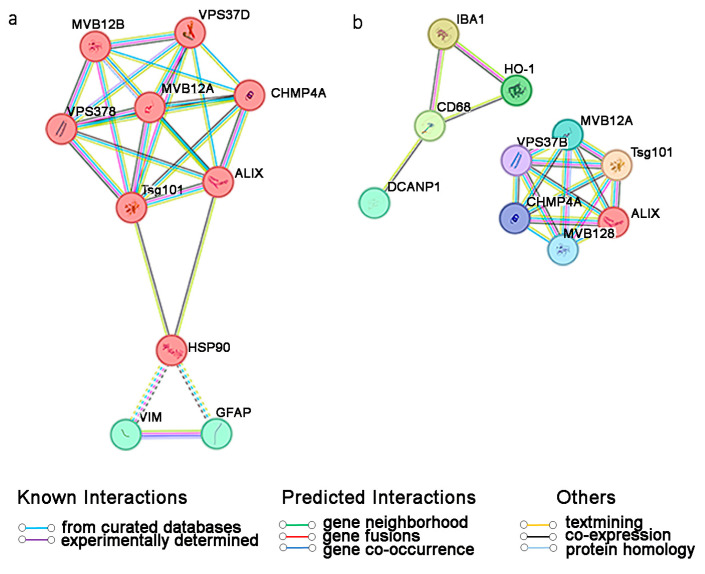
Protein–protein interaction analysis for (**a**) astrocytes and (**b**) myeloid cells with EVs targets. Each node represents all the proteins produced by a single, protein-coding gene locus. Colored, filled nodes represent query proteins and the first shell of interactors. The dotted lines indicate the protein clusters. Minimum required interaction score, medium confidence = 0.400.

**Figure 3 brainsci-14-00331-f003:**
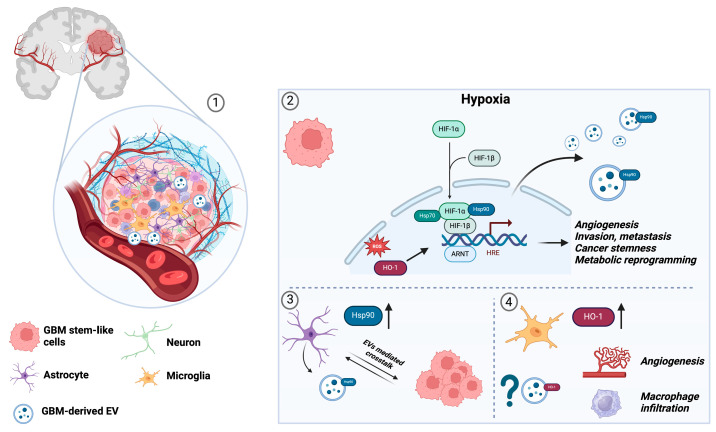
Schematic representation illustrating the role of extracellular HSP90 and HO-1 in the crosstalk between different cell types present in the tumor microenvironment of GBM. The illustration provides insight into the intricate interactions between astrocytes and microglia/macrophages within the tumor microenvironment (TME), highlighting their bidirectional communication facilitated by the release of extracellular vesicles (EVs). These cellular interactions have been extensively documented in the context of their association with tumor cells in the TME (1). In GSCs in hypoxic conditions, HIF-α recruits Hsp90 and Hsp70 in the cytoplasm and interacts with the importin for nuclear translocation. Nucleated HIF-α further recruits other cofactors such as HIF-b and p300/CBP and initiates gene transcription and hypoxia signaling. Hypoxia triggers the activation of numerous downstream target genes, among which is HO-1, which plays an antiapoptotic role by inducing mitogen-activated protein kinase pathways (2). Notably, the figure highlights specific connections identified between human astrocytes and HSP90, suggesting a pivotal role for HSP90 in orchestrating the trafficking of human astrocytic EVs. HSP90 serves as a crucial regulator of cancer cell growth and survival, and its heightened expression in pathological reactive astrocytes characterized by GFAP+ and VIM+ expression significantly contributes to tumor progression. Understanding the role of HSP90 in EV trafficking sheds light on a potential mechanism through which astrocytes modulate the TME and influence tumor behavior (3). Additionally, the figure depicts the expression of the HO-1 protein by microglia/macrophages, which serves as a marker of neovascularization. Importantly, HO-1 may also be released into the TME via EVs, indicating a possible contribution to the complex interplay between immune cells and the tumor microenvironment. This suggests a multifaceted role for EV-mediated communication in shaping the TME and influencing various aspects of tumor progression, including angiogenesis and immune regulation (4). The depicted crosstalk underscores the dynamic nature of cellular interactions within the TME and highlights the potential significance of targeting HSP90 and HO-1 pathways for therapeutic intervention in glioblastoma and other malignancies. (Image was created with BioRender.com online software: https://www.biorender.com. Accessed on 17 March 2024).

**Table 1 brainsci-14-00331-t001:** Lineage-expression proteins for human glioma cells, astrocytes, and myeloid cells. The symbol + indicates protein overexpression; the symbol - indicates protein underexpression, with the main corresponding references in the literature. Both symbols are shown in the case of variability.

Proteins	Glioma Cells	Astrocytes	Microglia	Ref.
GFAP	GFAPδ+	GFAPα+		[23,25]
VIM	+	+/-		[26]
SLC1A2		-		[25]
SLC1A3		-		[25]
CD63	+	-	-	[37]
Cx43	+/-	+		[22]
GLT-1	-	+		[28,29]
Iba1	-		+	[44]
CD68			+	[44]
C16			+	[44]
CD163			+	[44]

## Data Availability

The data presented in the study are openly available at https://string-db.org/.

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
