# Peer review of "The Interplay between Glioblastoma Cells and Tumor Microenvironment: New Perspectives for Early Diagnosis and Targeted Cancer Therapy"

_brainsci, 2024, doi:10.3390/brainsci14040331_

Round 1

Reviewer 1 Report

Comments and Suggestions for Authors

The manuscript "The interplay between glioblastoma cells and tumor microenvironment: new perspectives for early diagnosis and targeted cancer therapy" by Virtuoso et al. is an interesting review on the fastest advancements in GBM targeted therapies.

In the first lines of the introduction, authors say that chemotherapy and surgery are the only therapeutic options. However, there are other therapies being used in GBM, at least at an experimental stage as CAR-T cells and immunotherapies. Of note on March 2024 Bagley et al published an outstanding paper on CAR-T cells in GBM patients (Intrathecal bivalent CAR T cells targeting EGFR and IL13Rα2 in recurrent glioblastoma: phase 1 trial interim results by Bagley et al, Nat Med 2024). I recommend the authors to discuss this recent advances.

Also, radiation therapy deserves at least a brief mention, maybe particularly along with temozolomide as it is usually used in GBM patients with deep tumors and well tolerated.

Being a review, I don't know if the bioinformatics analysis should be included. More importantly, I think the analysis falls a little short, not showing any novelty nor shedding light on unknown pathways or implicated molecules. The analysis might well be removed, and the review is still very interesting. If it's included in the final version of the manuscript, the quality of figure 2 should be improved, since it gets pixelated when zoomed in.  Also, I think the bioinformatic analysis should have a materials and methods section instead of describing the analysis performed in the main text, but the editor surely can clarify this. Additionally, the links provided are expired. 

Comments on the Quality of English Language

The manuscript is well written, and I have only detected some minor English errors. Authors should re-read the manuscript.

Author Response

To

Mrs Jasmine Lu, Section Managing Editor of Brain Science

Dr. Stephen Meriney, Editor in Chief of Brain Science

Palermo, March 26, 2024

Dear Editors,

Thank you very much for your interest in our manuscript and for the opportunity to review it, as suggested by the Reviewers. We also thank the Reviewers for their very useful comments and suggestions, which we have followed in preparing the revised manuscript. In the revised version of the manuscript, the changes have been implemented using the "track changes" mode for visibility. Additionally, we have addressed the requested modifications to the references list, although these adjustments have not been explicitly highlighted to maintain the readability of the bibliography.

Below this message, you will find a point-by-point response to the Reviewers’ comments. We hope that this revised version of our paper is now satisfactory and meets the requirements for publication in Brain Sciences.

Sincerely,

Celeste Caruso Bavisotto

Celeste Caruso Bavisotto

Department of Biomedicine, Neuroscience and Advanced Diagnostics (BIND),

University of Palermo,

via del Vespro 129,

90127 Palermo, Italy

[email protected].                          

Reviewer 1:

The manuscript "The interplay between glioblastoma cells and tumor microenvironment: new perspectives for early diagnosis and targeted cancer therapy" by Virtuoso et al. is an interesting review on the fastest advancements in GBM targeted therapies.

Author response: We thank Reviewer for the constructive criticism, which will enhance the comprehensiveness and relevance of our manuscript.

R1 Comment 1. In the first lines of the introduction, authors say that chemotherapy and surgery are the only therapeutic options. However, there are other therapies being used in GBM, at least at an experimental stage as CAR-T cells and immunotherapies. Of note on March 2024 Bagley et al published an outstanding paper on CAR-T cells in GBM patients (Intrathecal bivalent CAR T cells targeting EGFR and IL13Rα2 in recurrent glioblastoma: phase 1 trial interim results by Bagley et al, Nat Med 2024). I recommend the authors to discuss this recent advances.

Response to comment 1. Thank you to the reviewer for this suggestion. We have modified the sentence in lines 39-40 of the Introduction section and the suggested reference was included in the References list (as reference number 6).

R1 Comment 2. Also, radiation therapy deserves at least a brief mention, maybe particularly along with temozolomide as it is usually used in GBM patients with deep tumors and well tolerated.

Response to comment 2. Thank you to the reviewer for this suggestion. We have included a brief mention of radiotherapy in lines 41-46 of the Introduction section and references were included in the References list (as reference numbers 3,7,8).

R1 Comment 3. Being a review, I don't know if the bioinformatics analysis should be included. More importantly, I think the analysis falls a little short, not showing any novelty nor shedding light on unknown pathways or implicated molecules. The analysis might well be removed, and the review is still very interesting. If it's included in the final version of the manuscript, the quality of figure 2 should be improved, since it gets pixelated when zoomed in.  Also, I think the bioinformatic analysis should have a materials and methods section instead of describing the analysis performed in the main text, but the editor surely can clarify this. Additionally, the links provided are expired. 

Response to comment 3. We appreciated the reviewer's comment. We included the investigation in the final version of the manuscript. We introduced a “methods” section, describing the performed investigation. We discussed the investigation in a new section entitled “New perspectives for early diagnosis and targeted cancer therapy” (please see lines 396-411 and 413-441, respectively).

The current investigation highlights perspectives and opinions for targeted studies and therapies, which are not found in other recent reviews. Based on the known protein-protein interactions, the STRING database sheds light on astrocytes and microglia/macrophage specificity regarding the HSP90 and HO-1 pathways, respectively. Moreover, HSP90 could be a target for activated-astrocytes-derived EVs, while studies on human microglia/macrophage EVs characterization are still limited.

We improved the quality of Figure 2. We deleted the expired links. However, we mentioned all the keywords and parameters for reproducing the investigation.

R1 Comment 4. The manuscript is well written, and I have only detected some minor English errors. Authors should re-read the manuscript.

Response to comment 4. The minor English errors were corrected.

Reviewer 2 Report

Comments and Suggestions for Authors

The review by Virtuoso et al on the topic of interplay between glioblastoma cells and tumor microenvironment: new perspectives for early diagnosis and targeted cancer therapy looks interesting. In the current review, they had presented the glioblastoma tumor microenvironment in context of astrocytes, microglia/macrophage, and extracellular vesicles. Although there have been already many articles and reviews published in this domain, however, authors had tried to summarize them in clear way.

I have carefully gone through the literature and context of the review and references provided in this paper. Here, I have some comments on the study:

In the introduction part (Line 47-51), authors had discussed about the neuroimaging techniques (MRI and CT) for the detection of gliomas. Can authors also have opinion on diffused gliomas and recurrent GBM, where these techniques failed to predict the exact size of the invasion on core of the tumor? https://doi.org/10.3174/ajnr.A8094

Typo in line 118 (Extracellular vesicles)

Can authors again visit ref – 35 (Glia, 2016. 64(8): p. 1416-36.10.1002/glia.23014) to understand the better context?

In figure 1, remove unnecessary “and” (in M2)

Line 194 (typo ‘Anti-tumor’)

In section 4 (Extracellular biomarkers involved in early diagnosis of GBM), Authors have mentioned (ref-60). I am agreeing on the context of liquid biopsy are somewhat provide the information about mutational load on GBM, however, this method is still in very early phase to claim the understanding of change in phenotype in tumor microenvironment because of infiltrative nature and subtype specific characters if each individual GBM patients.

I would suggest, if authors can provide information about progression free survival (PFS)/overall survival (OS) in the context of astrocytes and microglia and their influence in tumor microenvironment in GBM.

Can authors re-address ref-77?

In ref-79, authors may include specifically CCT6 (chaperonin containing TCP1 subunit 6A) in context of GBM.

For the figure 2, can authors include a short method in figure legend explaining median cutoff, confidence? (Links are not working provided in figure section)

 In figure 2, authors may include p-value to make strong evidence to the claim. The connections between HSP90 and EV have been mostly reported in the context of ubiquitin dependent endocytosis. Authors could also include this in the description.

Author Response

To

Mrs Jasmine Lu, Section Managing Editor of Brain Science

Dr. Stephen Meriney, Editor in Chief of Brain Science

Palermo, March 26, 2024

Dear Editors,

Thank you very much for your interest in our manuscript and for the opportunity to review it, as suggested by the Reviewers. We also thank the Reviewers for their very useful comments and suggestions, which we have followed in preparing the revised manuscript. In the revised version of the manuscript, the changes have been implemented using the "track changes" mode for visibility. Additionally, we have addressed the requested modifications to the references list, although these adjustments have not been explicitly highlighted to maintain the readability of the bibliography.

Below this message, you will find a point-by-point response to the Reviewers’ comments. We hope that this revised version of our paper is now satisfactory and meets the requirements for publication in Brain Sciences.

Sincerely,

Celeste Caruso Bavisotto

Celeste Caruso Bavisotto

Department of Biomedicine, Neuroscience and Advanced Diagnostics (BIND),

University of Palermo,

via del Vespro 129,

90127 Palermo, Italy

[email protected].                          

Reviewer 2:

The review by Virtuoso et al on the topic of interplay between glioblastoma cells and tumor microenvironment: new perspectives for early diagnosis and targeted cancer therapy looks interesting. In the current review, they had presented the glioblastoma tumor microenvironment in context of astrocytes, microglia/macrophage, and extracellular vesicles. Although there have been already many articles and reviews published in this domain, however, authors had tried to summarize them in clear way.

Author response: We thank Reviewer for positive comment.

I have carefully gone through the literature and context of the review and references provided in this paper. Here, I have some comments on the study:

R2 Comment 1. In the introduction part (Line 47-51), authors had discussed about the neuroimaging techniques (MRI and CT) for the detection of gliomas. Can authors also have opinion on diffused gliomas and recurrent GBM, where these techniques failed to predict the exact size of the invasion on core of the tumor? https://doi.org/10.3174/ajnr.A8094

Response to comment 1. We complemented the MRI with the current opinions on T2-FLAIR sequence and we included the reference suggested by the reviewer (please see lines 58-62)

R2 Comment 2. Typo in line 118 (Extracellular vesicles)

Response to comment 2. Typo was corrected.

R2 Comment 3. Can authors again visit ref – 35 (Glia, 2016. 64(8): p. 1416-36.10.1002/glia.23014) to understand the better context?

Response to comment 3. We apologize for the mistake. We changed the ref Glia, 2016. 64(8): p. 1416-36.10.1002/glia.23014 with a more appropriate reference: Marino, S., Menna, G., Di Bonaventura, R., Lisi, L., Mattogno, P., Figà, F., Bilgin, L., D'Alessandris, Q. G., Olivi, A., & Della Pepa, G. M. (2023). The Extracellular Matrix in Glioblastomas: A Glance at Its Structural Modifications in Shaping the Tumoral Microenvironment-A Systematic Review. Cancers, 15(6), 1879. https://doi.org/10.3390/cancers15061879.

R2 Comment 4. In figure 1, remove unnecessary “and” (in M2)

Response to comment 4. The figure was modified as the reviewer suggested.

R2 Comment 5. Line 194 (typo ‘Anti-tumor’)

Response to comment 5. Typo was corrected.

R2 Comment 6. In section 4 (Extracellular biomarkers involved in early diagnosis of GBM), Authors have mentioned (ref-60). I am agreeing on the context of liquid biopsy are somewhat provide the information about mutational load on GBM, however, this method is still in very early phase to claim the understanding of change in phenotype in tumor microenvironment because of infiltrative nature and subtype specific characters if each individual GBM patients.

I would suggest, if authors can provide information about progression free survival (PFS)/overall survival (OS) in the context of astrocytes and microglia and their influence in tumor microenvironment in GBM.

Response to comment 6. We agree that the use of liquid biopsy, including the assessment of circulating tumor cells (CTCs), is still in its early stages for understanding the dynamic changes in the tumor microenvironment (TME) in GBM. While we acknowledge the limitations in fully elucidating the impact of infiltrative nature and subtype-specific characteristics of individual GBM patients using liquid biopsy alone, we believe it offers valuable insights into mutational load and potential biomarkers for early diagnosis.

Regarding the suggestion to include information on progression-free survival (PFS) and overall survival (OS) in the context of astrocytes and microglia influence on the TME in GBM, we appreciate the importance of these clinical endpoints in assessing the prognostic significance of cellular interactions within the TME. We will revise the manuscript to include relevant studies that investigate the correlation between astrocytes, microglia, as suggested by the reviewer (Please see line 206-220).

R2 Comment 7. Can authors re-address ref-77?

Response to comment 7. We re-addressed reference 77 (now is ref. 86) in the manuscript and ensure that it is appropriately cited and discussed in the context of our research (Please see line 285-290).

R2 Comment 8. In ref-79, authors may include specifically CCT6 (chaperonin containing TCP1 subunit 6A) in context of GBM.

Response to comment 8. Thank you for your suggestion. We included specific mention of CCT6.

R2 Comment 9. For the figure 2, can authors include a short method in figure legend explaining median cutoff, confidence?

Response to comment 9. We added the median confidence level in the legend of Figure 2.

R2 Comment 10. In figure 2, authors may include p-value to make strong evidence to the claim.

Response to comment 10. We included the p-value levels in the text, as reported by STRING database (please see lines 421 and 468).

R2 Comment 11. The connections between HSP90 and EV have been mostly reported in the context of ubiquitin dependent endocytosis. Authors could also include this in the description.

Response to comment 11. We mentioned the connection between HSP90 and EVs as suggested by the reviewer (please see lines 367-378).
